Living to the range limit: consumer isotopic variation increases with environmental stress

Reddin Carl J. 1 creddin01@qub.ac.uk
O’Connor Nessa E. 2
http://orcid.org/0000-0002-5353-1556 Harrod Chris 3
1 Faculté des Sciences et des Techniques, Université de Nantes , Nantes , France
2 School of Biological Sciences, Queen’s University Belfast , Belfast , United Kingdom
3 Instituto de Ciencias Naturales Alexander Von Humboldt, Universidad de Antofagasta , Antofagasta , Chile
Reimer James
Electronic publication date: 2016 Jun 1
Publication date: 2016
Volume: 4
Electronic Location ID: e2034
Received 2016 Mar 12; Accepted 2016 Apr 22
Copyright: © 2016 Reddin et al.
Copyright year: 2016
Copyright holder: Reddin et al.
License: This is an open access article distributed under the terms of the Creative Commons Attribution License, which permits unrestricted use, distribution, reproduction and adaptation in any medium and for any purpose provided that it is properly attributed. For attribution, the original author(s), title, publication source (PeerJ) and either DOI or URL of the article must be cited.
License URL: https://creativecommons.org/licenses/by/4.0/

Keywords: Stress gradient, Intertidal, Range limit, Littorinid, Stable isotopes, Trophic niche width, Physiological condition, Diet, Trophic variation, Isotopic niche

Funding: CONICYT PAI MEL 81105006 CH and CR were funded by the CONICYT Grant PAI MEL 81105006. The funders had no role in study design, data collection and analysis, decision to publish, or preparation of the manuscript.

==============================
Background: Theoretically, each species’ ecological niche is phylogenetically-determined and expressed spatially as the species’ range. However, environmental stress gradients may directly or indirectly decrease individual performance, such that the precise process delimiting a species range may not be revealed simply by studying abundance patterns. In the intertidal habitat the vertical ranges of marine species may be constrained by their abilities to tolerate thermal and desiccation stress, which may act directly or indirectly, the latter by limiting the availability of preferred trophic resources. Therefore, we expected individuals at greater shore heights to show greater variation in diet alongside lower indices of physiological condition.

Methods: We sampled the grazing gastropod Echinolittorina peruviana from the desert coastline of northern Chile at three shore heights, across eighteen regionally-representative shores. Stable isotope values (δ13C and δ15N) were extracted from E. peruviana and its putative food resources to estimate Bayesian ellipse area, carbon and nitrogen ranges and diet. Individual physiological condition was tracked by muscle % C and % N.

Results: There was an increase in isotopic variation at high shore levels, where E. peruviana’s preferred resource, tide-deposited particulate organic matter (POM), appeared to decrease in dietary contribution, and was expected to be less abundant. Both muscle % C and % N of individuals decreased with height on the shore.

Discussion: Individuals at higher stress levels appear to be less discriminating in diet, likely because of abiotic forcing, which decreases both consumer mobility and the availability of a preferred resource. Abiotic stress might be expected to increase trophic variation in other selective dietary generalist species. Where this coincides with a lower physiological condition may be a direct factor in setting their range limit.

Introduction

Environmental gradients, such as of temperature or moisture, are often associated with variation in a species’ abundance and biomass (Austin, 1985), constraining its spatial range (Normand et al., 2009; Sexton et al., 2009) and its potential contributions to ecosystem functioning (Cardinale, Nelson & Palmer, 2000; Duffy, 2002). A species’ spatial range can therefore expose the limits of its ecological niche (Sexton et al., 2009). However, most superficial indicators of species performance (e.g. abundance) do not help to identify whether a species’ range limit is set directly or indirectly by the environmental variable(s) in question (Somero, 2002; Sexton et al., 2009). Towards the extremes of a species’ environmental tolerance (cf. Shelford’s Law of Tolerance; Odum & Barrett, 2005), rates of biochemical reactions in individuals may be affected (Menge & Sutherland, 1987), and decreased individual performance may be observed as decreased physiological condition. An environmental gradient that is expected to produce a linear response in organism performance is the combined thermal and desiccation stress in intertidal ecosystems (Lewis, 1964; Somero, 2002).

Typically a species’ geographical range spans hundreds or thousands of kilometres, but vertical ranges may span only a few metres in the intertidal zone (Connell, 1971; Raffaelli & Hawkins, 2012), where species are often distributed in conspicuous belts (Stephenson & Stephenson, 1949; Ingólfsson, 2005; cf. Continuum Concept, e.g. Austin, 1985). This tractable environmental gradient, identified as one of four major global ecoclines (Whittaker, 1975), forms as organisms that are principally marine compromise their physical and biological tolerances across height on the shore (Connell, 1961; Davenport & Davenport, 2005), such that it can predict species richness and diversity (e.g. Connell, 1978; Zwerschke et al., 2013). On the high shore, marine organisms are subjected to extreme abiotic stress from the synergistic effects of several physical factors combined (Underwood, 1979; Menge & Sutherland, 1987; e.g. desiccation, temperature; Davenport & Davenport, 2005), while other physical factors may oppose one another (e.g. wave exposure vs. desiccation; Menge & Sutherland, 1987; Harley & Helmuth, 2003; Gilman, 2006). The effect is that no single factor (e.g. elevation) drives the stress gradient, and integrated stress indicators, such as species dominance patterns, may be more useful in defining the gradient (Menge & Sutherland, 1987).

Components of a species’ habitat can be quantitatively represented as geometric dimensions, brought together as potential niche space in a multi-dimensional hypervolume (Hutchinson, 1978). A species’ theoretical optimum can be represented as a centroid or kernel in this niche space, with an individual suffering decreased performance when it is located far from this optimum (Whittaker, Levin & Root, 1973; Odum & Barrett, 2005). A unimodal response in organism performance might then be expected around the optimum (fundamental niche). This Hutchinson (1978) concept of niche space is applied in stable isotope ecology, where isotopic axes represent n-space axes (Bearhop et al., 2004; Newsome et al., 2007), with δ13C and δ15N typically representing variation in food web baseline (i.e. primary producers) and consumer trophic position, respectively. In particular, isotopic variance among individuals in the population can be informative regarding an organism’s isotopic niche width, a close proxy for trophic niche width (Bearhop et al., 2004; Newsome et al., 2007; Fink et al., 2012). Innovative methods based on ecomorphological techniques allow the quantification and comparison of isotopic niche width (Layman et al., 2007; Jackson et al., 2011; Syväranta et al., 2013). Additionally, the percentages of muscle C and N can provide proxies for physiological condition, with a decline in % C indicating a decline in stores of lipids and carbohydrates, and a decline in % N indicating a decline in tissue protein content (Nalepa et al., 1993).

Environmental stress gradients may be associated with changes in resource use; it might be expected that under ideal conditions a generalist but selective species may specialise on its preferred resource (Harrod, Mallela & Kahilainen, 2010), reducing niche width. However, environmental stress may decrease the abundance of the preferred resource or limit an organism’s ability to forage (Hidalgo et al., 2008). This may encourage the assimilation of previously discounted resources (Harrod, Mallela & Kahilainen, 2010). Individuals may then be expected to either generalise their resource use similarly across the population (all feeding on equal proportions of different resources) or each individual to specialise their resource use variously (each feeding on distinct resources; Valen, 1965; Bearhop et al., 2004). Individual specialisation appears to be widespread among animal species (Bolnick et al., 2003; Araújo, Bolnick & Layman, 2011), and the degree of dietary specialisation of consumers has proven important in governing the strength of top-down control (Duffy et al., 2007). Still, few studies have calculated the magnitude of individual specialisation under different contexts (Araújo, Bolnick & Layman, 2011), such as a species’ range limit.

We estimated the δ15N, δ13C, % N and % C of representative individuals of the regionally common grazing gastropod Echinolittorina peruviana (Lamarck, 1822) and its putative food sources. Individuals were sampled at three shore height levels at several rocky shores in northern Chile. Based on current theory (Hidalgo et al., 2008), we anticipated that individuals on the high shore, which approach the upper limit of this species distribution and a more harsh physical environment than the lower shore, would be forced to limit their foraging and only consume food resources that are immediately available. Littorinid snails are ubiquitous in the upper intertidal such that, historically, they were recommended for general definition of the supralittoral limit on shores worldwide (Stephenson & Stephenson, 1949; Lewis, 1964). Furthermore, littorinids are generalist but selective grazers, having a feeding preference for easily digested microalgae and ephemeral macroalgae (Lubchenco, 1978; Norton et al., 1990). We predicted that: (1) the isotopic niche width of E. peruviana will be broader in populations inhabiting the high shore compared to those lower on the shore (individuals forced into consuming distinct resources); and (2) the physiological condition indices of muscle % C and % N, representing proportion of carbon stores (lipids and carbohydrates) and protein content respectively (e.g. Nalepa et al., 1993), will be lower in high shore individuals relative to low shore individuals.

Materials and Methods

Study region

The Humboldt Current system intertidal is well described (Santelices, Vasquez & Meneses, 1986; Fernández et al., 2000; Broitman et al., 2001), typically with dense beds of the mussel Perumytilus purpuratus on the mid shore (Broitman et al., 2001). Our study was conducted on the Mejillones Peninsula of northern Chile, close to the city of Antofagasta (approx. 23°20′S, 70°30′W). On this coastline that forms the western extreme of the Atacama Desert, tidal emergence brings numerous disadvantages for intertidal organisms, including extremely high solar intensity (Antofagasta summer monthly average direct normal solar radiation = ~7000 Wh/m2; EnergyPlus, 2013), extremely low rainfall (Antofagasta mean = 7 mm year−1; Climatemps, 2013) and warm air temperature (summer monthly mean = 20.1 ± 0.1 °C; EnergyPlus, 2013). The Mejillones Peninsula exemplifies strong variation in coastal upwelling, induced by coastal topography, and associated spatial variation in intertidal assemblage structure (Marín, Delgado & Escribano, 2003; Thiel et al., 2007; Reddin et al., 2015). Reddin et al. (2015) found E. peruviana abundance to have a strong negative association (r = −0.74, P < 0.001) with a proxy for the intertidal influence of upwelling. The upper shore is mainly dominated by E. peruviana (average density, 57 ± 99 individuals m−2), chthamaloid barnacles and Ulva spp (Nielsen & Navarrete, 2004). The Antofagasta Bay intertidal is also inhabited by the invasive ascidian Pyura praeputialis, which was a dominant intertidal bioengineer, covering exposed rocks and increasing local invertebrate diversity (Castilla et al., 2004).

Field sampling

Because the absolute height and width of species’ vertical ranges (i.e. that of E. peruviana) may vary corresponding to local conditions (e.g. Stephenson & Stephenson, 1949), we use the emergent biological feature of organism vertical zonation to define levels of integrated thermal and desiccation stress (Menge & Sutherland, 1987). Therefore, E. peruviana individuals were collected from the following zones: high shore shallow pools (< 10 cm) above all emergent macroalgae (high); above barnacle/Perumytilus zone (middle); and within or below barnacle/Perumytilus zone (low). E. peruviana individuals were not found above these zones. Hidalgo et al. (2008) found an average vertical distance of 0.35 m between these ‘mid’ and ‘low’ shore heights at the lower latitude of 11°46′S, in Peru.

E. peruviana individuals were sampled concurrently with Reddin et al. (2015), during February to March 2012, at the same sites (n = 17, Euclidean distance between the most northern and southern sites = ~83 km), which represented regional coastline variation (see Reddin et al., 2015, for sites detail). Similar target sized individuals were collected (~12.5 mm, longest axis), aiming to minimise ontogenetic variation, and then pooled into low, mid and high groups (Table 1). Although there remained size variation among sampled individuals, this was not significantly different among the three shore height groups (one-way ANOVA for length, MS = 0.04, F1,58 = 0.02, P = 0.89). Shore height groups were balanced to the smallest group size (n = 20) by randomly dropping surplus individuals per height per site (i.e. maintaining regional representation).

Table 1 Mass and length (mean ± SD) of E. peruviana individuals at three relative shore height groups.

Height	Mass (g)	Length (mm)	Samples (n)	
Low	0.3 ± 0.1	10.9 ± 1.4	20	
Mid	0.3 ± 0.1	10.9 ± 1.1	20	
High	0.3 ± 0.1	11.0 ± 1.5	20	

We sampled the following putative trophic resources at all locations where E. peruviana were sampled: Ulva sp., brown macroalgae (either Dictyota sp. or Lessonia nigrescens; fresh growth only), epilithic biofilm (scraped from rocks using a spatula) and particulate organic matter (POM; a 5 l water sample filtered through a 0.7 μm Whatman GF/F filter until the filter showed a colour change). The only situation where littorinids had apparent access to all putative resources was on the low shore. At all heights, we predicted that POM residue from outgoing tides would remain available as a resource but at diminishing quantities with increased shore height; it would be doubtful that the other resources would occur in a thriving state. Biofilm would be present at a far thinner thickness on the mid-shore, and probably of a different flora, while macroalgal matter could drift in, including to the high shore. No other consistent and obvious resources were present at mid- and high shore heights.

We recorded individual littorinid wet-mass (in shell, to 0.1 g; mass range 0.1–0.7 g) and length (longest axis; to 0.1 mm), then dissected the muscle tissue (Lorrain et al., 2002) and washed it with distilled water. Fresh macroalgal growth was isolated and rinsed with distilled water. POM samples were filtered through pre-combusted (550 °C for 5 h) 0.7-μm Whatman GF/F filters. All samples were oven-dried at 65 °C for 48 h to a constant mass. Animal, biofilm and macroalgal samples were then homogenised and weighed (± 0.01 mg) into tin capsules (6 × 4 mm, Sercon Ltd) on a Mettler Toledo XS3DU Microbalance, or cut into sections (GF/F filters). Biofilm samples were duplicated, with one duplicate prepared as above for δ15N and the other decalcified to remove inorganic calcium carbonate contaminants (Ng, Wai & Williams, 2007). Decalcification proceeded by application of 10% hydrochloric acid drop by drop until bubbling ceased (Carabel et al., 2006), after which samples were re-homogenised and standardised by dry mass into tin capsules as above.

Samples were combusted in a continuous-flow elemental analyser (CHNOS) interfaced to a mass spectrometer (IsoPrime 100, Center for Stable Isotope Biogeochemistry, University of California at Berkeley) to estimate δ13C, δ15N, and elemental % C and % N values. Isotope ratio data were expressed in the standard δ unit, as the ratio of heavy to light isotopes, in ‰ units: δ (‰) = [(Rsample/Rreference) − 1] × 103, with R = 13C/12C for carbon and 15N/14N for nitrogen. Reference materials for the above calculation were the international standards of V-PDB for C and air for N. We used two calibration standards: the external standard ‘peach leaves,’ NIST SMR 1547, showed analytical precision to be 0.10 and 0.15 ‰ for δ13C and δ15N, and an internal standard, Patella vulgata muscle, suggested precision to be < 0.1 ‰ for both δ13C and δ15N.

Statistical analyses

We estimated isotopic niche width (δ13C and δ15N) by Bayesian ellipses (SEA.B), and also estimated carbon range (CR) and nitrogen range (NR) separately using a Bayesian implementation of the metrics by Layman et al. (2007; all SEAc, SEA.B, CR and NR from R package SIBER; Jackson et al., 2011). CR and NR split the bivariate isotopic niche represented by the SEA.B into its univariate δ13C and δ15N axes. All estimates were calculated by 104 posterior density draws and are reported using mode and 95% credible intervals. Statistical significance of differences between pairs of groups was estimated by comparing proportional overlap of posterior densities. We also tested for any interaction of vertical variation in isotopic niche width or physiology with oceanographic variation identified by Reddin et al. (2015), by geographically splitting individuals sampled from Antofagasta Bay (sites and Mejillones Peninsula (again, replicates balanced between geographical groups’ shore height levels, by randomly dropping surplus replicates) and re-calculating SEA.B).

Monotonic relationships were tested for using Spearman’s rank correlations, with 95% confidence intervals calculated using the R package ‘psychometric’ (Fletcher, 2010). We compared standard correlation significance values with those accounting for spatial autocorrelation via the Dutilleul et al. (1993, in R package ‘Spatial Pack,’ Osorio & Vallejos, 2014) method, which adjusts the degrees of freedom appropriately where spatial non-independence of data is detected. In no case did this change the level of significance so, for simplicity, we present the standard test results.

To estimate proportional contribution of the sampled putative resources to the diet of E. peruviana, we used the Bayesian mixing model SIAR (Parnell et al., 2010), with individuals split by shore height. We used the trophic fractionation values for the intertidal mollusc Mytilus edulis (Dubois et al., 2007; δ13C = 2.2 ± 0.1, δ15N = 3.8 ± 0.1). To facilitate model runs, we performed a number of steps: (a) concentration dependence of resources was included; (b) macroalgal species, in particular Ulva sp., had overlapping δ13C values, and so data were grouped a priori to model runs; (c) to represent geographical variation in the intertidal isotopic baseline, resources and consumers were split into Antofagasta Bay and Mejillones Peninsula following Reddin et al. (2015). Still, model runs frequently yielded large credible intervals for estimated resource contributions (particularly for the high shore) so we only report cases of significant differences in the same shore height group (within or between the Bay and Peninsula), as estimated by comparing proportional overlap of posterior densities. All analyses were implemented within the R statistical package (R Development Core Team, 2008).

Results

Isotopic niche widths showed a marked positive trend with shore height (Fig. 1), with low shore individuals being isotopically most similar (SEA.B mode = 2.47 ‰2, 95% credible intervals = 1.61–3.88; significantly smaller than mid-shore SEA.B, probability of difference > 99.99%), mid-shore individuals having intermediate variation (mode 11 ‰2, 95% CI = 7.21–17.5; significantly smaller than high shore, probability > 95%) and high shore individuals being isotopically most dissimilar (SEA.B over eight times larger than that of low shore individuals; mode 21.6 ‰2, 95% CI = 13.9–33.4; significantly larger than low shore SEA.B, probability > 99.99%). This positive trend was shown for both C and N ranges (Fig. 2), with N range increasing most sharply, although credible intervals were large for these metrics (differences between shore heights were insignificant; probability < 95%). The positive trend between shore and SEA.B was observed over both Antofagasta Bay and Mejillones Peninsula, appearing more severe over the Mejillones Peninsula (Fig. 3) although no shore height group was significantly different in SEA.B between the bay and the peninsula. However, Mejillones Peninsula had significantly lower densities of E. peruviana (18 ± 57 individuals m−2) relative to Antofagasta Bay (88 ± 115 individuals m−2; P < 0.01, t-test).

Figure 1 Isotopic variation among Echinolittorina peruviana individuals grouped by relative height on the shore.

(A) Individual values on an δ15N-δ13C biplot, with standard ellipses (equivalent to a bivariate SD), corrected for small sample size (SEAc), plotted as calculated by the SIBER procedure (Jackson et al., 2011). Plot lines and symbols represent shore height (see legend). (B) E. peruviana isotopic niche width (SEA.B) at different shore heights, plotted with 95, 75 and 50% credible intervals and mode as a black point. Macroalgae includes Lessonia nigrescens, Dictyota sp. and Ulva sp. pooled.

Figure 2 Separate (A) carbon and (B) nitrogen isotopic variation of E. peruviana individuals from different shore heights.

Range estimates calculated by a Bayesian implementation of Layman et al. (2007) in the R package SIBER (Jackson et al., 2011). A black dot shows the mode, while boxes represent the 95, 75 and 50% credible intervals.

Figure 3 Comparison of isotopic niche widths (SEA.B) of E. peruviana individuals, grouped by relative height on shore, over Antofagasta Bay (black boxes) and Mejillones Peninsula (white boxes).

This geographical split follows the ecological differences identified over this region by Reddin et al. (2015). Plotted are 95, 75 and 50% credible intervals (boxes) and the mode (black or white dot). Sample sizes per height level for both geographical groups, ‘low’ n = 10, ‘mid’ n = 10, ‘high’ n = 7.

We identified negative correlations between relative shore height and both individual muscle % C and % N (Fig. 4; % carbon rs = −0.34, df = 58, P < 0.01, 95% confidence intervals = −0.55–−0.10; % nitrogen rs = −0.52, df = 58, P < 0.0001, 95% CI = −0.69–−0.31), with a proportionally greater drop in % N than % C. This led to an increase in C:N ratio with increasing relative shore height (Fig. 4C; rs = 0.51, df = 58, P < 0.0001; 95% CI = 0.31–0.69).

Figure 4 Trend of decreasing (A) % C and (B) % N, and (C) increasing C:N, of E. peruviana muscle tissue with increasing relative shore height.

Shown are the median (dark line), inter-quartile range (box), range (whiskers) and outliers (points).

Mixing model derived dietary contribution estimates were most precise for low shore individuals (Fig. 5), having narrower credible intervals, while mid- and high shore contribution estimates overlapped considerably. On the low shore, POM (Bay mode = 0.59, 95% credible intervals = 0.41–0.73; Peninsula mode = 0.39, 95% CI = 0.22–0.53) contributed consistently more than biofilm (Bay mode = 0.25, 95% CI = 0.03–0.4; Peninsula mode = 0.15, 95% CI = 0.02–0.28; Fig. 5) in both bay and peninsula (both probability > 99%), while macroalgae (mode = 0.43, 95% CI = 0.24–0.73) contributed significantly more than biofilm (mode = 0.15, 95% CI = 0.02–0.28) on the peninsula (probability > 95%) and in the bay macroalgae contributed (mode = 0.1, 95% CI = 0–0.43) less than POM (mode = 0.59, 95% CI = 0.41–0.73; probability > 95%). On the mid shore, contribution estimates of biofilm were most precise, both Antofagasta Bay and Mejillones Peninsula having low modal values (Bay mode = 0.27, 95% CI = 0.01–0.46; Peninsula mode = 0.06, 95% CI = 0–0.33, lower than POM, probability = 94%), although differences between estimates for all resources were insignificant within mid- and high shore levels. Macroalgal contribution estimates were higher in peninsula locations than in the bay, for both low and mid-shores, but large credible intervals made this difference insignificant at the 5% level (probability = 94% between low shore estimates).

Figure 5 Contributions of putative resources to E. peruviana diet at high (A and B), middle (C and D) and low (E and F) shore heights.

Contributions estimated by SIAR mixing models (Parnell et al., 2010) run separately for Antofagasta Bay (A, C, E) and Mejillones Peninsula (B, D, F). Plotted are contribution estimates’ 95, 75 and 50% Bayesian credible intervals (boxes), mode values (dots) and probability of differences between putative resource contributions of POM, macroalgae (L. nigrescens, Dictyota sp. and Ulva sp. pooled), and epilithic biofilm.

Discussion

We found isotopic niche widths of an intertidal grazer to be significantly larger approaching the limit of its vertical distribution, coinciding with significant decreases in % C and % N, both candidate proxies for individual physiological condition (e.g. Nalepa et al., 1993). Vertical shore height is a well understood stress gradient for marine organisms (Lewis, 1964; Valdivia et al., 2011) and governs their vertical distribution (Stephenson & Stephenson, 1949; Somero, 2002). We document a case where individuals approaching their upper distribution limit, which is set by abiotic factors, may exhibit a more varied, suboptimal, potentially indiscriminating resource use, supported by a decrease in physiological condition and isotope mixing models. We compare evidence for this conclusion and for competing explanations of an enlarged isotopic niche width.

A large population isotopic niche width for this generalist species suggests that individuals within this population were isotopically distinct, and likely consumed isotopically distinct foods. Conversely, isotopically more similar individuals, showing lower isotopic variation at the population level, likely consumed isotopically similar (on average) foods (Cummings et al., 2012). The positive association between population isotopic niche width and putative abiotic stress therefore supports our hypothesis = that individuals at the high shore extreme of their environmental tolerance were forced to individually specialise their diet. E. peruviana, as well as other high shore littorinids, preferentially feeds on the epilithic microbial film (Hawkins et al., 1989; Norton et al., 1990; Mak & Williams, 1999; Hidalgo et al., 2008), including bacteria, microalgae, macroalgal spores and early growth stages (Norton et al., 1990; Vermeulen et al., 2011). E. peruviana can also consume crustose, foliose and filamentous macroalgae (Santelices, Vasquez & Meneses, 1986). We used the isotopic composition of muscle tissue, which has a turnover rate in marine molluscs of between 3–6 months (Lorrain et al., 2002). The isotopic composition of the assimilated resource is therefore averaged over that period, during which the mobile snail could move considerably, potentially even between the stress zones of this study, resembling other mobile species approaching their distributional limits. Benthic grazers can move considerably per tide to optimise their foraging, up to 1.5 m for the European littorinid, Littorina littorea, (Newell, 1958) and tidal vertical migrations of up to 1 m for a tropical limpet (Williams & Morritt, 1995). Therefore, we suggest that exchange of individuals between low and mid shore heights may be frequent but between mid and high levels would be low. Hidalgo et al. (2008) investigated the grazing of E. peruviana on wave-sheltered Peruvian shores, using two shore heights corresponding to ‘low’ and ‘mid’ heights in our study. They found that E. peruviana could control epilithic biofilm abundance better on the low shore and suggested that feeding was limited at greater height on the shore by faster desiccation of the rock’s surface, decreasing the window for foraging per tidal cycle (also Mak & Williams, 1999; Norton et al., 1990). Additionally, greater quantities of mucus required to aid locomotion was suggested to burden the gastropod’s energy reserves (Calow, 1974; Hidalgo et al., 2008). Littorinids have been reported to have higher rates of radular activity at higher shore locations, and higher shore species to feed more rapidly than low shore species (Newell, Pye & Ahsanullah, 1971). Foraging movement limitation may mean that littorinids in high shore habitats have to be more flexible in their diet, reflecting reduced resource availability. For instance, field observations of high-shore E. peruviana clustered over and leaving grazing scars on washed up kelp stipes support occasional consumption of macroalgae (see also Voltolina & Sacchi, 1990). In contrast, lower shore individuals may be able to move around and be more selective in their diet. Individual specialisation is widespread among animal species (Bolnick et al., 2003), including aquatic gastropods (Doi et al., 2010, using stable isotope analysis), although it has not been associated with approaching a species’ spatial range limit.

We found both E. peruviana muscle % C and % N concentrations to decrease significantly with increasing putative stress. A relative decrease in % C is often interpreted as a decrease in tissue lipid content or carbohydrate storage (Nalepa et al., 1993), such as observed in freshwater molluscs during reproduction (Aldridge, 1982, via a decrease in C:N) or during low food availability (Russell-Hunter, Browne & Aldridge, 1984). Physiological processes resulting in the loss of muscle elementary N include high excretion of nitrogenous wastes, which increase with individual size, higher temperatures or longer durations, in the fish Salmo trutta (Elliott, 1976). Intertidal snails, however, need water to excrete their ammonia or urea. Protein catabolism was found to be higher in a high shore littorinid species relative to a low shore species (Littorina saxatilis and Littorina obtusata, respectively), but both had their highest rate of protein catabolism at lower temperatures (Aldridge, Russell-Hunter & McMahon, 1995). Protein catabolism and nitrogenous excretion could explain our observed % N decrease, but % N was not significantly related to δ15N in our samples (rs = −0.17, P = 0.21). Our individuals were size-matched, but both average temperature and the duration of emergence were expected to increase with height on the shore (Muñoz et al., 2008). High shore individuals were only found associated with small rock pools, presumably because of the shelter afforded from thermal and desiccation stress. For instance, Williams & Morritt (1995) recorded individual limpet body temperatures from high shore shallow pools on Hong Kong Island to be consistently ~2 °C cooler than exposed nearby limpets, although they noted that these pools could also become hypersaline or dry out after tidal emergence. Accordingly, Hidalgo et al. (2008) found that E. peruviana individuals inhabiting the high-shore were smaller, potentially due to less favourable conditions, although size-selective predation on the lower shore could also explain this. Whether our high shore individuals were there by intention or not is unclear but it is likely that the observed (by proxy) lower physiological condition might incur a lower fitness in high shore individuals, concordant with range limit conditions (Sexton et al., 2009).

We interpret our mixing model results to suggest that POM was the preferred food of E. peruviana, most probably that deposited on the rock during tidal immersion, which, we propose, becomes more scarce at greater shore height. The Humboldt region is characterised by high pelagic production (Thiel et al., 2007), which subsidises intertidal food webs (Reddin et al., 2015). E. peruviana was concluded by Hidalgo et al. (2008) to exert top-down control on biofilm as well as on crustose macroalgae, but had little effect on larger turf and canopy macroalgae. We suggest that our samples of biofilm, which were scraped to the underlying rock, may represent a resource that was too thick for the small littorinid’s radula to fully penetrate, and that its grazing would have focussed on the surface layers of biofilm. A biofilm, especially with diatoms, can facilitate the deposition of suspended particles (i.e. POM) by production of an extracellular polymeric secretion (EPS; Decho, 2000). At higher shore heights, shorter tidal immersion time and greater UV radiation leads to less abundant biofilm (Thompson, Norton & Hawkins, 2004), itself leading to less deposited and captured POM. Together, combined with a shorter foraging time as the rocks dry during tidal emersion, could decrease E. peruviana’s consumption of preferred POM. Additionally, we observed an insignificant difference in rate of increase in isotopic variation with greater shore height, between Antofagasta Bay and Mejillones Peninsula. Higher POM concentrations in the waters of Antofagasta Bay (Reddin et al., 2015) may increase POM deposited with each tide, supporting high shore individuals. The higher abundance of ecosystem engineers in Antofagasta Bay (P. purpuratus and Pyura praeputialis; Castilla et al., 2004; Reddin et al., 2015; although see Castilla et al., 2014) could also ameliorate mid-intertidal conditions. Animals are expected to choose food resources that minimise energetic costs, including finding, handling and digesting the resource, thereby yielding the maximum ‘value’ per unit metabolic cost (Townsend & Hughes, 1981). This can be aided by behavioural maintenance of the animal’s proximity to resources (Underwood, 1979).

The translation of isotopic niche to the trophic niche can be complicated by physiological conditions in the consumer or the resource(s). Stress caused by thermal extremes or water loss could itself contribute to the isotopic variation observed in high shore individuals. Intertidal organisms, including littorinids, have evolved behavioural and physiological tolerance mechanisms for greater desiccation and insolation (Aldridge, Russell-Hunter & McMahon, 1995; Somero, 2002; Muñoz et al., 2008). Some of these physiological conditions can affect δ15N: excretion of nitrogenous waste in the form of urea, typical for higher intertidal littorinids (e.g. Littorina saxatilis; Aldridge, Russell-Hunter & McMahon, 1995), is associated with consumer enrichment in 15N (Vanderklift & Ponsard, 2003); starvation or nutritional stress can result in elevated values of δ15N (Vanderklift & Ponsard, 2003; Hobson, Alisauskas & Clark, 2013). A comparably high isotopic variation has nevertheless been observed previously in a grazing gastropod (the freshwater species Lymnaea stagnalis; Doi et al., 2010; δ13C range 11.6 ‰, δ15N range 7.4 ‰). Despite this variation being large relative to other sampled consumers, Doi et al. (2010) concluded that variation in resource use underpinned isotopic variation. Environmental temperature can affect trophic fractionation from resource to ectothermic consumer (Power, Guiguer & Barton, 2003; Barnes et al., 2007). Atmospheric temperature is not expected to vary across the sites but rock surface temperatures could vary depending on orientation to the sun, colour and type of rock. Water temperature was expected to differ by ~3 °C between bay and outer peninsula situations (Reddin et al., 2015) but the resultant difference in trophic enrichment (+0.3 ‰ for both 13C and 15N, extrapolated from Barnes et al., 2007) was anticipated to be negligible (e.g. level of analytical error ± 0.1 ‰). We cannot rule out isotopic variation in the sources (Flaherty & Ben-David, 2010; Cummings et al., 2012); even if our consumer was feeding on the same resource(s) across shore heights a broader population isotopic niche width could result from variation in the basal resource(s). Thompson, Norton & Hawkins (2004) investigated biofilm biomass and diatom abundance at two different UK shore heights. They found strong negative correlations of biomass and diatom abundance with insolation stress and air temperature, but not with grazing intensity or dissolved nutrients (Thompson, Norton & Hawkins, 2004). Therefore, the fine-scale spatial variation of thermal and desiccation stress in the intertidal are likely to have a strong influence on biofilm biological and isotopic composition (Vermeulen et al., 2011), although this may be partially alleviated by the spatial averaging effect provided by mobile consumers (Post, 2002; Vermeulen et al., 2011). Cummings et al. (2012) recommend that the confounding effect of source variation is eliminated by transforming isotopic niche width to trophic niche width, by converting delta space to dietary proportions (p-space; Newsome et al., 2007). We were precluded from fully utilising this step, particularly on the high shore, because of difficulties identifying high shore resources and knowledge of the effect of environmental stresses at this level on isotopic variation.

When we combined our % C and % N values into the ratio C:N, we observed an increase in C:N ratio with increasing putative environmental stress via shore height. Increasing C:N is commonly used as a proxy for increasing organism condition, especially by lipid concentration (e.g. Schmidt et al., 2003), but such use has been criticised (Kidd et al., 2011). For example, beside condition, individual lipid content can vary with tissue type and stage in life history (e.g. reproductive stages). Furthermore, Kidd et al. (2011) found the relationship between C:N and lipid content to vary significantly among populations, with predictive models performing best within populations. An alternative interpretation of increasing C:N is a relative decrease in N, the variation of which can exceed that of C (Kidd et al., 2011), as was the case in our study. Ultimately, a ratio is the relationship between two numbers; in the case of C:N, neither % C nor % N is solely dependent on the other, and an increase in the ratio can be derived from a change in either term. Where C:N is used as a proxy for physiological condition, we recommend that researchers support its appropriateness by presenting % N and % C, especially if processes that could change tissue % N are anticipated in their study system.

Environmental factors can induce limits on the effective feeding time of consumers, depending on their adaptations; for instance, ice or snow can impede grazing or predation in the high Arctic (Connell, 1971). Range limits, whether set by altitude in mountainous terrain, height on the intertidal or latitude across a continent, remain the spatial expression of the ecological niche (Sexton et al., 2009), suggesting that the upscaling of our results to other taxa, realms and stress scenarios may be possible. The optimal spatial scale of study depends on the acuteness of the stress gradient, but where exact replication is unlikely, variation in background processes should be integrated in the replication. This allows the averaging-out of the effects of alternative sympatric stress gradients (e.g. wave exposure in our study; see Valdivia et al., 2011) that may synergise or antagonise the effects of the focal stress gradient; thus, the conclusions develop generality. Therefore, the study extent, either spatial or temporal, must cover variation in regional (or seasonal) sources of stress, but not be so large that new food resources are introduced that could falsely inflate measures of trophic variation (Araújo, Bolnick & Layman, 2011). Conveniently, stable isotope ratios of organism tissues naturally provide a temporal (and spatial, depending on a species’ mobility) averaging effect (Post, 2002), although the large foraging areas of highly mobile consumers may demand the adoption of large spatial grains.

Supplemental Information

Supplemental Information 1 C & N concentration for bay mixing model.

Click here for additional data file.

Supplemental Information 2 Source isotopic values for bay mixing model.

Click here for additional data file.

Supplemental Information 3 C & N concentration for headland mixing model.

Click here for additional data file.

Supplemental Information 4 Source isotopes for headland mixing model.

Click here for additional data file.

Supplemental Information 5 Main data table.

Click here for additional data file.

We thank Joel Trexler for useful comments on an earlier version of the manuscript, two anonymous reviewers for comments that improved it further, and Felipe Docmac for help with sample collection.

Additional Information and Declarations

Competing Interests

Author Contributions

Data Deposition

The authors declare that they have no competing interests.

Carl J. Reddin conceived and designed the experiments, performed the experiments, analyzed the data, wrote the paper, prepared figures and/or tables, reviewed drafts of the paper.

Nessa E. O’Connor conceived and designed the experiments, wrote the paper, reviewed drafts of the paper.

Chris Harrod conceived and designed the experiments, performed the experiments, analyzed the data, contributed reagents/materials/analysis tools, wrote the paper, reviewed drafts of the paper.

The following information was supplied regarding data availability:

The raw data has been supplied as Supplemental Dataset Files.

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
