# Peer review of "Living to the range limit: consumer isotopic variation increases with environmental stress"

_PeerJ, doi:10.7717/peerj.2034_

## Round 0.1 · original submission · Minor Revisions

I have heard back now from two reviewers. Both were generally positive about your manuscript, and both offered constructive and helpful comments. As well, reviewer 1 has asked for some more information and clarification. These reviewers' comments will help you improve your manuscript, but I do not foresee major changes in it; therefore, my decision is 'minor revision'.

Reviewer 1 ·

Basic reporting

Review of Reddin et al.

This manuscript investigates the change of diets of Echinolittorinids along the tidal levels from high, mid to low shores, using a stable isotope approach. The samplings of the food sources, snails and analysis of the data is up to standard. The MS can be accepted for publication after revisions. There are some basic assumption and biology of the species needed to be clarified in details in the introduction and methods:
1) How is the foraging range of the snails in high, mid and low shores. Mobile gastropods can have great distance moved and can forage among tidal levels. The author should state evidence that the Echonilittorind snails they studied does not have had among tidal level movements.
2) In the high shore, the author also sampled snails from rock pools. Rock pool water interface is a relatively benign environment (See WILLIAMS, G.A. & MORRITT, D. 1995. Habitat partitioning and thermal tolerance in a tropical limpet, Cellana grata on a tropical rocky shores. Marine Ecology Progress Series 124: 89-103). This should represent a separate group rather than grouped in the high shore for analysis?
3) Any seasonal variation in the algal diversity and biofilm diversity on the shores. In Hong Kong, grazers are exposed to different sources in algae in summer and winter, which can greatly influence the stable isotope value of the grazers.
4) The author say their study species is a generalist but selective grazer. What does it mean, does it mean it has feeding preference on certain types of algae? This need elborations.
5) What is the species in the macroalgae samples for food source analysis? Any examination of the species in the biofilm samples?

Experimental design

Good. comments are in the basic reporting.

Validity of the findings

Important.

Additional comments

In the basic reporting.

Reviewer 2 ·

Basic reporting

No comments

Experimental design

No Comments

Validity of the findings

No Comments

Additional comments

The manuscript is very well written and clear. I see no major flaws in the study. The authors should be aware that Harrod et al. may soon publish some work refuting the hypothesis that isotopic and trophic niche size are proportional... this work was recently presented at a Stable Isotope Ecology Conference... however, these data are not yet published. Some minor comments:

I feel that “credible interval” is more widely used than “credibility interval” when referencing Bayesian statistics.

Odd character —— long dashes in lines 209 and 210.

The authors should explain the mechanism for reduced % C and % N to indicate poor physiological condition. Since these two factors co-varied, it seems that the C/N ratio is constant and therefore not a very informative measure of variance in body composition. Indeed, this is discussed in the Discussion section but the C:N ratios are not reported or shown in a figure. I suggest making figure 4 a 3 panel figure with C/N as well.

Line 286 has an unrecognized character where a delta symbol should be.

---

## Round 0.2 · accepted · Accept

The manuscript is revised well, and is now acceptable for publication. Please note that I have added two small corrections to the paper; please see the attachment and edit in the proofs or an earlier stage (if possible).